# Lyme disease and relapsing fever in Mexico: An overview of human and wildlife infections

**Pablo Colunga-Salas**[1,2], **Sokani Sánchez-Montes**[1], **Patricia Volkow**[3], **Adriana Ruíz-Remigio**[1], **Ingeborg Becker**[1] *

1 Centro de Medicina Tropical, Unidad de Investigación en Medicina Experimental, Facultad de Medicina, Universidad Nacional Autónoma de México, Ciudad de México, México, 2 Posgrado en Ciencias Biomédicas, Unidad de Posgrado, Universidad Nacional Autónoma de México, Ciudad de México, México, 3 Departamento de Infectología, Instituto Nacional de Cancerología, Ciudad de México, México

* becker@unam.mx

**Data Availability Statement:** All relevant data are within the paper and its Supporting Information files.

## Abstract

Lyme borreliosis and Relapsing fever are considered emerging and re-emerging diseases that cause major public health problems in endemic countries. Epidemiology and geographical distribution of these diseases are documented in the US and in Europe, yet in Mexico, studies are scarce and scattered. The aims of this study were (1) to present the first confirmatory evidence of an endemic case of Lyme disease in Mexico and (2) to analyze the epidemiological trend of these both diseases by compiling all the information published on *Borrelia* in Mexico. Two databases were compiled, one of human cases and another of wild and domestic animals in the country. The analysis included the evaluation of risk factors for the human population, the diversity of *Borrelia* species and their geographic distribution. Six *Borrelia* species were reported in a total of 1,347 reports, of which 398 were of humans. Women and children from rural communities were shown to be more susceptible for both Lyme borreliosis and Relapsing fever. The remaining reports were made in diverse mammalian species and ticks. A total of 17 mammalian species and 14 tick species were recorded as hosts for this bacterial genus. It is noteworthy that records of *Borrelia* were only made in 18 of the 32 states, mainly in northern and central Mexico. These results highlight the importance of performing further studies in areas where animal cases have been reported, yet no human studies have been done, in order to complete the epidemiological panorama for Lyme borreliosis and Relapsing fever. Finally, the search for *Borrelia* infections in other vertebrates, such as reptiles and amphibians is recommended to gain a more accurate view of *Borrelia* species and their distribution. The geographical approach presented herein justifies an intense sampling effort to improve epidemiological knowledge of these diseases to aid vector control and prevention programs.

## Introduction

Tick-borne pathogens (TBP) have become a public health problem due to the continuous rise in the incidence of human and animal diseases associated with TBPs [1]. Some TBPs that can

**Funding:** Our study received grant from Universidad Nacional Autonoma de Mexico (UNAM) with grant number: PAPIIT IN211418. The funders had no role in study design, data collection and analysis, decision to publish, or preparation of the manuscript. Pablo Colunga-Salas received a fellowship from CONACyT with number 463798.

**Competing interests:** The authors have declared that no competing interests exist.

affect both humans and animals include several bacterial members of the genus *Borrelia* [1–3]. This bacterial genus comprises various spirochetal Gram-negative species that are divided into four phylogenetic groups: reptile-associated *Borrelia* (REP), monotreme-associated *Borrelia* (MAB), relapsing fever (RF) and Lyme-borreliosis (LB) [4–6]. The pathogenic species for humans of this bacterial genus are found only in the RF and LB groups [2, 5, 7].

Taxonomically, species of the genus *Borrelia* were initially considered as part of the genus *Spirochaeta*, but in 1907 the genus was divided and the genus *Borrelia* was formally described [8]. For more than a century, there were no important taxonomic changes. However in 2014, phylogenomic and protein studies led to the proposal of creating a new genus for members of the LB group (*Borreliella*) [9]. Further studies on the percentage of conserved proteins were proposed as a suitable method to delimitate the bacterial genera and the genus *Borrelia* was again proposed as a monophyletic genus [10]. To date, controversy remains on whether the genus should be divided [11, 12].

The *Borrelia* species causing RF are mainly transmitted by soft argasid ticks of the genus *Ornithodoros* and the human body louse *Pediculus humanus*, even though hard ticks of the genera *Ixodes*, *Amblyomma* and *Rhipicephalus* have also been implicated as potential vectors [13–15]. To date, 27 *Borrelia* species and two *Candidatus* species have been described as members of this monophyletic group [3, 15]. Of these, 24 and one *Candidatus* species have been recognized as pathogens for humans and/or other mammalian orders (Table 1) [3].

The *Borrelia* species that cause LB are grouped into 22 species [16], which are clustered into a monophyletic group. Of these, 11 species have been found to be the etiological agents of Lyme borreliosis, also known as Lyme disease (Table 1) [2, 17–20]. The competent vectors of these bacterial species are hard ixodid ticks [2, 17].

In Mexico, information on the diversity of *Borrelia* species, of wildlife infections and of human cases remains controversial [55–57]. The current information on this genus is poorly scattered and often published in local bulletins lacking diffusion. Furthermore, no confirmatory evidence of endemic human cases of Lyme disease has been shown in Mexico. Therefore, new insights and more solid evidence of human and animal infections with *Borrelia* are needed to elaborate an actualized map of these bacteria in Mexico.

Thus, the aim of this study was to: A) present to first confirmatory report of an endemic human case of Lyme disease in Mexico, and B) compile all the published records of human and animal infections by *Borrelia* in Mexico, to give an accurate picture of the current epidemiological situation of the genus *Borrelia* in the country.

## A) Case report

### Clinical summary and test procedures

A 67 years-old, unemployed male patient sought medical attention in August 2017. He was born in the coastal state of Veracruz, Mexico, where his family has a cattle farm. He had a previous history of Brucellosis at age 38. Diabetes had been diagnosed 15 years prior to the current office visit and had been irregularly controlled. He was hypertensive and treated with amlodipine, valsartan and hydrochlorothiazide. He had a history of multiple trips to the State of Veracruz, during which he had seen cattle infested with ticks. After visiting the family farm in the northern part of Veracruz in August 2016, he developed a red skin lesion that lasted several weeks, yet the patient did not seek medical attention. He started having diplopia and fatigue in September 2016. This evolved to stabbing headaches, cramps and pain in his limbs, bilateral hypoesthesia in hands and feet (in gloves and socks) and tachycardia. After having lost 15 Kg, he was hospitalized in March 2017, showing distal strength decrease +/+++ in lower limbs and sensitivity decrease in all modalities (exteroceptive and proprioceptive). A lumbar puncture

**Table 1. Pathogenic *Borrelia* species.**

| *Borrelia* group | Species | Region/Country | Disease | References |
|---|---|---|---|---|
| RF | *Borrelia baltazardii* | Iran | TBRF (Tick-borne relapsing fever) | [21] |
| | *Borrelia braziliensis* | Brazil | TBRF | [15, 22] |
| | *Borrelia caucasica* | Caucasus area | TBRF | [23] |
| | *Borrelia coriaceae* | Western North America | Bacteremia of deer | [15, 24] |
| | *Borrelia crocidurae* | Western and northern Africa | TBRF, mild symptoms | [3, 15, 25] |
| | *Borrelia dugesii* | Mexico | TBRF | [26] |
| | *Borrelia duttonii* | Central, eastern and southern Africa | TBRF, Neurological signs, neonatal infections | [3, 15, 27] |
| | *Borrelia graingeri* | Kenya | Flu-like syndrome | [28, 29] |
| | *Borrelia harveyi* | Kenya | Bacteremia of monkeys | [15, 30] |
| | *Borrelia hermsii* | Western North USA, Brirish Columbia (Canada) | TBRF | [15, 31, 32] |
| | *Borrelia hispanica* | Iberian Peninsula and northern Africa | TBRF | [15, 33] |
| | *Borrelia latyschewii* | Central Asia and Middle East | TBRF, Flu-like syndrome | [15, 34] |
| | *Borrelia lonestari* | Southern and eastern United States | Bacteremia of deer | [15, 35] |
| | *Borrelia mazzottii* | Mexico, Central America and Western USA | TBRF | [15, 26] |
| | *Borrelia microti* | Iran | TBRF | [15, 36] |
| | *Borrelia miyamotoi* | Europe, Asia and North America | TBRF, Flu-like syndrome | [15, 37] |
| | *Borrelia parkeri* | Western USA | TBRF | [15, 32] |
| | *Borrelia persica* | Central Asia, Middle East, Egypt and India | TBRF | [15, 38] |
| | *Borrelia queenslandica* | Australia | Bacteremia with relapse in mice | [3] |
| | *Borrelia recurrentis* | Africa (Global)* | Louse-borne relapsing fever | [3, 39] |
| | *Borrelia theileri* | Africa (Global)** | Bovine borreliosis | [15, 40] |
| | *Borrelia turicatae* | British Columbia (Canada), Southwestern and south-central United States and Mexico | TBRF | [15, 41] |
| | *Borrelia venezuelensis* | Central America and northern South America | TBRF | [3, 42] |
| | *Candidatus* Borrelia kalaharica | Africa | TBRF | [43, 44] |
| LB | *Borrelia afzelii* | Europe and Asia | Lyme Disease (LD) | [45] |
| | *Borrelia americana* | North America | LD | [46] |
| | *Borrelia andersonii* | US | LD | [47] |
| | *Borrelia bavariensis* | Europe | Lyme borreliosis | [48] |
| | *Borrelia bissettii* | North America and Europe | LD | [49] |
| | *Borrelia burgdorferi* s.s. | East and West United States and Eastern Europe | LD | [50] |
| | *Borrelia garinii* | Europe and Asia | LD | [51] |
| | *Borrelia lusitaniae* | Mediterranean basin | LD | [52] |
| | *Borrelia mayonii* | Upper midwestern US | Lyme borreliosis | [20] |
| | *Borrelia spielmanii* | Europe | LD | [53] |
| | *Borrelia valaisiana* | Europe and Japan | LD | [54] |

* *B. recurrentis* human cases have been reported in Ethiopia and Sudan, however a worldwide distribution is suspected.

** *B. theileri* cases have been reported in Africa, Australia, North and South America, but due to global bovine trade, it is now considered to be globally distributed.

was performed, and the spinal fluid showed xanthochromia, glucose 69%, proteins 305 mg/ml, 98 mononuclear and 2 polymorphic nuclear cells (PMN)/μl. He received treatment with ceftriaxone and prednisolone (PDN) for two weeks. The headache remitted, the remaining symptoms improved by 70% and he was able to walk and climb stairs.

In August 2017, he again sought medical attention because symptoms restarted, he reported fatigue, diplopia and bilateral hypoesthesia in hands and feet (in gloves and socks), that made walking clumsy and difficult, fine tremor in his hands and short-term memory loss. On physical exam there was generalized areflexia, no meningeal signs and diminished strength in both lower limbs. He was again hospitalized and blood (5 ml) was taken in 2 vacutainer tubes, one with and one without ethylenediaminetetraacetic acid (EDTA). The sera samples were centrifuged at 2,000 rpm during 10 min and stored at -20°C until use. DNA of whole blood samples was extracted by columns (DNEasy Blood and Tissue Kit, QIAGEN Inc., DEU), quantified with a spectrophotometer (Nanodrop-1000, Thermo Fisher, USA) and adjusted to a final concentration of 300 ng. Serological analysis was done by Western blotting using anti-*Borrelia burgdorferi*-WESTERNBLOT IgM and IgG kits (EUROIMMUN, Medizinische Labordiagnostik AG, D-23560 Lübeck, DEU), following instructions of the manufacturer. For molecular detection of *Borrelia*, a ~280 bp fragment of the *flagellin* protein gen (*fla*) was amplified by conventional PCR [58]. The reaction mixture consisted of 12.5 µl GoTaq® Green Master Mix, 2X (Promega Corporation, Madison, WI, USA), 2µM of each pair of primers, 6.5 µl nuclease-free water and 50 ng DNA in a final volume of 25 µl [58]. Negative and positive controls were included. As positive control, DNA from *Borrelia* previously isolated from *Amblyomma dissimile* (Colunga-Salas et al. unpublished data, GenBank accession number KY389373) was used. PCR product was visualized in 2% agarose gels with SmartGlow™ Pre-Stain (Accuris Instruments, Edison, NJ, USA) and visualized by UV-transillumination.

A written informed consent was signed by the patient, who was informed of the publication of his case.

## Results

The serological analysis by Western blotting was positive for IgG and indeterminate for IgM. The PCR product showed a band of ~280 bp. Sequencing of the PCR product was done at Laboratorio de la Biodiversidad y la Salud, Instituto de Biología, Universidad Nacional Autónoma de México. The sequence was deposited in GenBank with the accession number MN607028. Molecular identification of the patient isolate was done by editing and aligning the sequences manually, including other species of the LB group. Bayesian analysis in MrBayes 3.2.3 [59] was done using the Markov Chain Monte Carlo (MCMC) algorithm with 10,000,000 generations and sampling every 1,000 generations, with a burning of 25% and the substitution model (Hasegawa, Kishino, and Yano model with gamma distribution [HKY+G] [60]) calculated in JModelTest 2 [61] based on the Bayesian Information Criterion. The convergence of the phylogenetic analysis was checked and considered as good when the ESS was higher than 200, in Tracer 1.7.1 [62].

The molecular confirmation by PCR showed that this patient was positive for *B. burgdorferi* s. s. (Fig 1), with a posterior probability of 0.96. The BLAST analysis showed 100% of identity and e value of 4e-133 with sequences of *Borrelia burgdorferi* s.s including those belonging to the strain B31 (Accession numbers: CP019767, AE000783, AB035617, X15661, L29200, AF416433 and Y15088) and other North American sequences. This result represents the first confirmed human autochthonous case of Lyme disease and includes the first available *Borrelia* sequence of Mexico.

## B) Compilation of *Borrelia* studies in Mexico from published records

### Methodology for databases

In addition to the written informed consent signed by the patient for the Case Report, the current study was approved by the Ethics and Research Committee of the Medical Faculty,

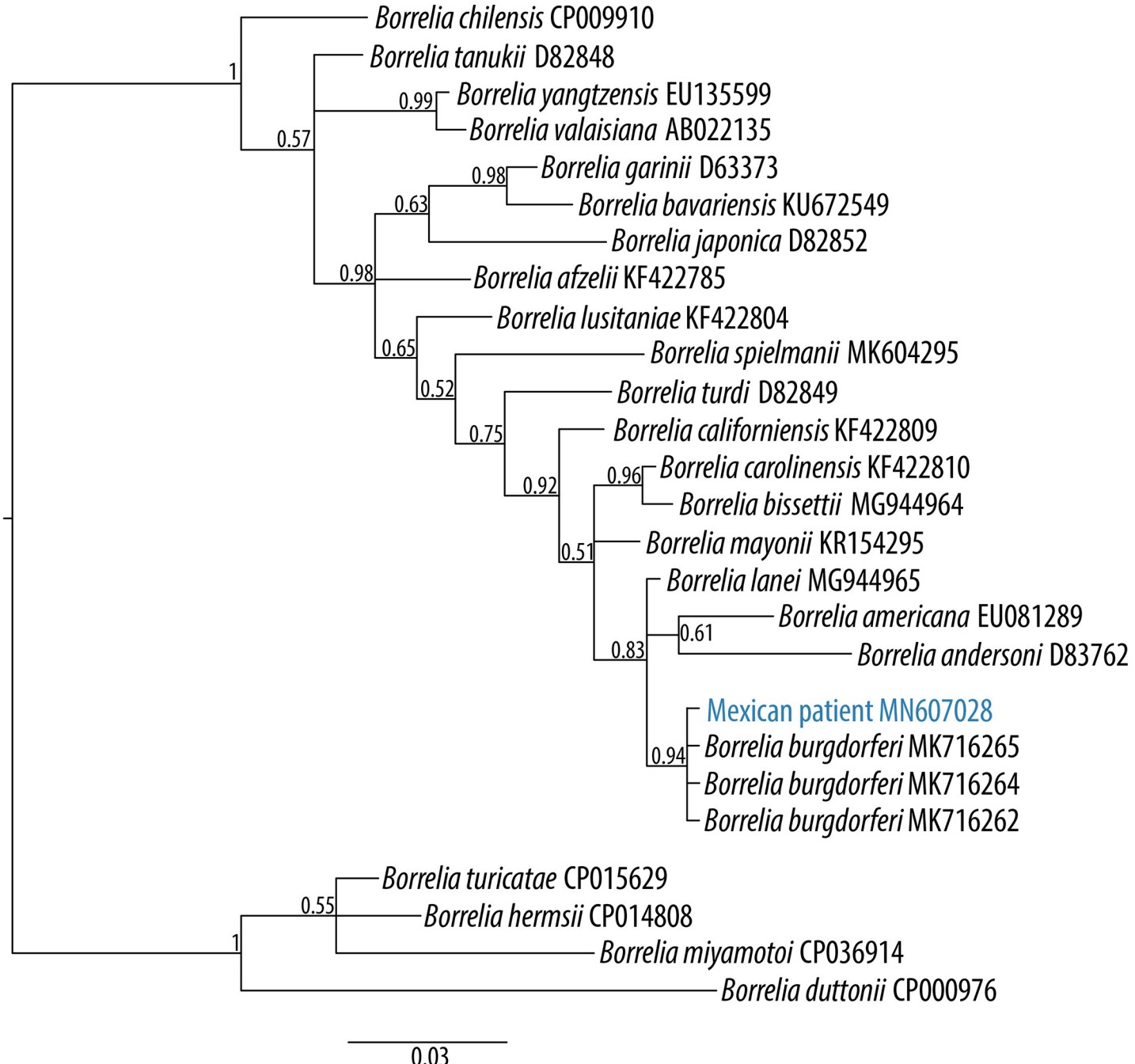

**Fig 1. Phylogenetic reconstruction for *flagellin* gene fragment of the Mexican patient with Lyme disease and of several members of the genus *Borrelia*.** The phylogenetic relationships were inferred by Bayesian Inference based on the HKY+G substitution model with a total of 255 bp. Posterior probabilities >0.5 are indicated at nodes. Number in parentheses are GenBank accession numbers. Scale bar indicates nucleotide substitutions per site. Blue sequence indicates isolate obtained from the recent Mexican patient.

UNAM (Comisiones de Investigación y de Ética de la División de Investigación de la Facultad de Medicina, UNAM), with approval number FM/DI/088/2017.

An extensive research of the literature was made to compile a database of published studies of *Borrelia* in human cases, mammals and ticks in Mexico from 1939 to 2020. A combination

of keywords: "Borrelia", "borreliosis", "mammals", "Mexico" and "human cases" were used in specialized databases including BioOne, Elsevier, Highwire, Iris, JSTOR, Pubmed, Scopus, SpringerLink, Wiley Online, Web of Science and Zoological Records, as has been previously proposed [63]. Articles reported in those studies were also analyzed by cross-referencing. Studies published in national and international journals were included.

**Human database.**   For the human database, the information was analyzed according to: (i) *Borrelia* species, (ii) number of human cases, (iii) sex, (iv) age group (newborn [0–2 yo], childhood [3–12 yo], teenager [13–18 yo], adult [19–60 yo] and elderly population [+60 yo]), (v) type of population (rural or urban), (vii) vector species, (viii) year of publication, (ix) detection method and (x) locality.

Additionally, information data from "National Health Information System [Sistema Nacional de Información en Salud] (SINAIS)" published by the Mexican Ministry of Health [Secretaría de Salud] [64] were obtained. This database includes all human borrelia-cases reported between 2000–2013 to the Ministry of Health. All the cases were georeferenced according to the "Catalogue of Keys from the Federal States, Municipalities and Localities" [Catálogo Único de Claves de Áreas Geoestadísticas Estatales, Municipales y Localidades] [65]. The SINAIS database did not include personal information of patients, nor the method of identification of the pathogens, for this reason, these records were only included for the geographical analysis.

**Animal database.**   From each study on mammals and/or ticks, the following information was recorded: (i) *Borrelia* species, (ii) number of positive animals for each species, (iii) order, family, genus and species of the mammalian host (if available), (iv) tick family, genus and species (if available), (v) year of collection, (vi) detection method and (vii) locality. Wild mammalian taxonomy was updated following the most recent taxonomical review for Mexico [66]. For ticks, a review for each genus was used [67–70].

**Database analyses.**   In order to present the first distribution map of *Borrelia* species, we generated a colorimetrical map in R, using the open access layers provided by CONABIO [Comisión Nacional para el Conocimiento y Uso de la Biodiversidad http://www.conabio.gob.mx/informacion/gis/] and functions from the R packages *viridis* [71], *tidyverse* [72], *maps* [73], *ggrepel* [74], *mapproj* [75] and *plotly* [76]. Maps were made for human cases, mammal and tick records, as well as a global map of total reports in Mexico, highlighting the zones with the highest number of cases, using the free and open source geographic information system, QGIS 2.18.9 [https://qgis.org/en/site/].

## Results of databases for *Borrelia* studies in Mexico

A total of 1,347 records were obtained from 39 published studies of *Borrelia* in Mexico. Of the total cases, 29.5% (398/1,347) corresponded to humans and 70.5% (949/1,347) were of other animal species (S1 and S2 Tables). Only one study reported both human and animal infections (S1 Table).

*Borrelia* species were specified in 54% (727) of the 1,347 cases. In the remaining 46% (620 cases), the etiological agent was identified as a member of the *B. burgdorferi* s.l. complex or as a member of the RF group. A total of six *Borrelia* species have been reported in Mexico [*B. afzelii*, *B. burgdorferi* s.s., *B. duguesii*, *B. garinii*, *B. mazzottiii* and *B. turicatae*] (Table 2), of which two are members of the LB group and four of RF.

The most frequent tests used for *Borrelia* detection were serological tests (Enzyme-Linked Immunosorbent Assay [ELISA], Western blotting and immunofluorescence assay [IFA]), used in 25 studies. This was followed by molecular tests (PCR) used in 11 studies. Five non-confirmed human cases were diagnosed by clinical manifestations and light microscopy.

**Table 2.** *Borrelia* species reported in Mexico.

| *Borrelia* group | *Borrelia* species | No. of records | Type of host |
|---|---|---|---|
| RF | *B. duguesii* | 1 | Animal |
| | *B. mazzottii* | 1 | Animal |
| | *B. turicatae* | 549 | Animal |
| LB | *B. afzelii* | 2 | Human |
| | *B. burgdorferi s.s.* | 128 | Human |
| | | 36 | Animal |
| | *B. garinii* | 10 | Human |

**Human cases.** In Mexico, 398 human cases of *Borrelia*-infection were recorded between 1939 and 2020. The most frequent diagnosis was LB in 98.7% (393/398) of the cases. RF was only diagnosed in 1.3% (5/398) of the human cases. Of the 393 cases with LB, in 35.6% (140/393) the etiological agent was identified (*B. afzelii*, *B. garrinii* and *B. burgdorferi* s.s.), using specific diagnosis tests (S1 Table). In contrast, no RF cases reported the infecting *Borrelia* species.

The gender of patient was specified in 231 (58%) of the 398 notified cases (S1 Table). The proportion between sexes was higher in females (134 cases) than males (97 cases). In patients with LB, women were the most frequent group with 58% (131/226), as compared to males showing 42% (95/226) of the cases (S1 Table). The remaining five patients corresponded to RF, which showed an almost equal distribution between sexes, with three cases being females and two cases were males.

The type of community of origin was referred only in 53.5% (213/398) of the cases (S1 Table). Of these 213 cases, 95.7% (204/213) corresponded to urban environments, the remaining 4.3% (9/213) were from rural areas. Human cases of RF showed that 80% (4/5) were reported in rural communities, whereas LB patients were mostly detected in urban communities 97.5% (203/208).

The age of patients was not specified in 94.4% (376/398) of all human cases, yet the remaining 22 cases (17.5%) corresponded mostly to adults 77.2% (17/22).

**Animal reports.** A total of 949 records of 17 mammalian species (nine wild species, three domesticated species, and two peri-domestic mammals) of six orders were reported as infected by *Borrelia* sp. in 19 studies from Mexico (Table 3; S2 Table). Four species of *Borrelia* were identified, three of which were members of the RF group (*B. duguesii*, *B. mazzottii* and *B. turicatae*) and only one from the LB group (*B. burgdorferi* s.s.) (S2 Table). The RF group was reported in 551 of the 949 records (58%), whereas members of the *B. burgdorferi* s.l. complex were reported only in 398 records [41.9%] (S2 Table).

The order Rodentia showed the highest number of infected species with nine rodent positive species, followed by carnivores with three (Table 3). In both of these mammalian orders, only the *B. burgdorferi* s.l. complex was recorded. In vertebrate hosts, the samples most frequently used for detecting *Borrelia* DNA was blood and/or tissues (305 records). Studies on the 305 mammalian samples only showed infection of an unknown species of the LB group.

Studies on arthropods and potential *Borrelia* vectors revealed 14 species: three soft ticks of the family Argasidae and 11 species of hard ticks of the family Ixodidae (Table 4), of which *I. scapularis* was the most common tick species positive for *Borrelia* DNA (Table 4; S2 Table). Studies on ticks as vectors were done in 570 records. Only 75 records were obtained from ticks retrieved from mammals (S2 Table).

Studies done with free-living ticks (569 records) showed that 96.7% (550/569) contained *Borrelia* causing RF, whereas only 19 records [3.3%] reported the LB group (S2 Table). The specificity of the *Borrelia* groups (only reported in ticks) showed that RF members were

**Table 3. Mammalian species associated to *Borrelia* in Mexico.**

| Mammalian species | | | Borrelia species |
|---|---|---|---|
| Order | Family | Species [English common name/Spanish common name] | |
| Artiodactyla | Cervidae | *Odocoileus virginianus* [White-tailed deer/Venado cola blanca] | *B. burgdorferi* s.l. |
| | Bovidae | *Bos taurus*\* [Aurochs /Toro] | *B. burgdorferi* s.l. |
| Carnivora | Canidae | *Canis lupus familiaris* [Dog/Perro] | *B. burgdorferi* s.l. |
| | | *Vulpes macrotis*\* [Kit Fox /Zorra del desierto] | *B. burgdorferi* s.s. |
| | Felidae | *Panthera onca*\* [Jaguar/Jaguar] | *B. burgdorferi* s.l. |
| | Procyonidae | *Bassariscus astutus*\* [Ringtail/Cacomixtle] | *B. burgdorferi* s.l. |
| | | | *B. burgdorferi* s.s. |
| Lagomorpha | Leporidae | *Sylvilagus floridanus*\* [Eastern Cottontail/Conejo] | *B. burgdorferi* s.s. |
| | | | *B. burgdorferi* s.l. |
| Perissodactyla | Equidae | *Equus caballus* [Horse/Caballo] | *B. burgdorferi* s.l. |
| Primates | Hominidae | *Homo sapiens*\* [Human/Humano] | *B. burgdorferi* s.s. |
| Rodentia | Cricetidae | *Microtus mexicanus* [Mexican Vole/Meteorito] | *B. burgdorferi* s.l. |
| | | *Neotoma mexicana* [Mexican Woodrat/Rata magueyera] | *B. burgdorferi* s.l. |
| | | *Neotoma micropus*\* [Southern Plains Woodrat/ Rata magueyera] | *B. duguesii* |
| | | *Neotomodon alstoni* [Volcano Deermouse/Ratón de los volcanes] | *B. burgdorferi* s.l. |
| | | *Peromyscus leucopus* [White-footed Deermouse/Ratón] | *B. burgdorferi* s.l. |
| | | *Peromyscus maniculatus* [North American Deermouse/Ratón] | *B. burgdorferi* s.l. |
| | Heteromyidae | *Heteromys pictus*\* [Painted Spiny Pocket Mouse/Ratón espinoso] | *B. burgdorferi* s.l. |
| | Muridae | *Mus musculus* [House Mouse/Ratón de casa] | *B. burgdorferi* s.l. |
| | | *Rattus rattus* [Roof Rat/Rata negra] | *B. burgdorferi* s.l. |

All wild species were updated according to the last taxonomic review of Ramírez-Pulido et al. [66]. Mammalian English common names were updated according to Wilson and Reeder [77] and Spanish common names according to Ceballos and Oliva [78].

\* In these studies, the authors did not include host samples when testing for *Borrelia* DNA.

recovered from argasid tick species, whereas the LB members were only detected in Ixodid tick species.

**Geographic analysis.** A total of 1,364 cases (both human and animal) were retrieved from published studies (1,347 cases) and from the National Health Information System (17 cases), of which 1,166 (85.4%) were geo-referred (at least at State level). These cases were distributed in 18 of the 32 Mexican states (Fig 2; S1 and S2 Tables). The state with most records was Aguascalientes (409/1,364 = 30%), followed by Nuevo Leon (253/1,364 = 18.5%) and Guanajuato (143/1,364 = 10.4%). The states with no records were Baja California Sur, Campeche, Chiapas, Colima, Durango, Guerrero, Hidalgo, Puebla, Queretaro, Tlaxcala and Zacatecas (Fig 2).

The highest diversity of *Borrelia* species was shown in the central and northeastern regions of the country, being Mexico City, the state with the largest number of species registered (3/6 = 66.7%), followed by Coahuila with two species [2/6 = 33.3%] (Fig 3).

The 1,166 geo-referred cases published both in the literature and by the National Health Information System corresponded mostly to animals 939 (80.5%) and only 19.4% (227) to human cases. Human cases were distributed in northeastern and central Mexico, mostly in the state of Nuevo Leon, where 69 of the 244 geo-referred human cases (28.2%) were reported with regard to their place of origin. This was followed by Tamaulipas with 18.8% (46/244) and Mexico City with 17.2% [42/244] (Fig 4).

*Borrelia*-infected animals have been reported in northeastern and northwestern regions, central and southeastern Mexico, with the highest density occurring the state of

**Table 4. Tick species recorded to be associated with *Borrelia* species in Mexico.**

| Tick species | | Mammalian species | *Borrelia* species |
|---|---|---|---|
| Family | Species | | |
| Argasidae | *Ornithodoros turicata* | ND | *B. turicatae* |
| | *Ornithodoros duguesi* | *Neotoma micropus* | *B. duguesii* |
| | *Ornithodoros talaje* | ND | *B. mazzottii* |
| Ixodidae | *Amblyomma americanum* | *Homo sapiens sapiens* | *B. burgdorferi* s.s. |
| | *Amblyomma cajennense* | *Bos Taurus* | *B. burgdorferi* s.s. |
| | | *Canis lupus familiaris* | *B. burgdorferi* s.s. |
| | | ND | *B. burgdorferi* s.s. |
| | *Amblyomma mixtum* | *Canis lupus familiaris* | *B. burgdorferi* s.l. |
| | *Dermacentor andersoni* | ND | *B. burgdorferi* s.s. |
| | *Dermacentor variabilis* | ND | *B. burgdorferi* s.l. |
| | *Ixodes affinis* | *Canis lupus familiaris* | *B. burgdorferi* s.l. |
| | *Ixodes kingi* | *Vulpes macrotis* | *B. burgdorferi* s.s. |
| | *Ixodes scapularis* | *Heteromys pictus* | *B. burgdorferi* s.l. |
| | | *Panthera onca* | *B. burgdorferi* s.l. |
| | | *Sylvilagus floridianus* | *B. burgdorferi* s.s. |
| | | *Sylvilagus floridianus* | *B. burgdorferi* s.l. |
| | | ND | *B. burgdorferi* s.l. |
| | *Ixodes spinipalpis* | ND | *B. burgdorferi* s.l. |
| | *Ixodes texanus* | *Basariscus astutus* | *B. burgdorferi* s.s. |
| | *Ixodes tovari* | ND | *B. burgdorferi* s.l. |
| | *Rhipicephalus sanguineus* s.l. | *Canis lupus familiaris* | *B. burgdorferi* s.l. |

Aguascalientes, that reported 406 of the 938 animals (43.2% with known place of origin), Nuevo Leon (184/938 = 19.6%) and Guanajuato (142/938 = 15.1%). The lowest density was reported in the central states of Mexico (Michoacan, Coahuila, Jalisco and San Luis Potosi) with less than 1% of the reports (Fig 5).

## Discussion

The first record of *Borrelia* in Mexico was reported by Brumpt [79], who recorded *B. turicatae* in specimens of the questing soft tick, *O. turicata*, collected in the states of Aguascalientes, Guanajuato and San Luis Potosi. The first report of *Borrelia* in humans was made by Pilz and Mooser [80], who described three cases of RF in Aguascalientes, based on microscopic evidence of spirochetes in thick-film samples and on symptoms of the patients [80].

It was not until the 1990s when the first record of human exposure to LB was published in Mexico by Maradiaga-Ceceña et al. [81], who reported 32 humans seropositive to strains from the LB group in Sinaloa, yet these studies did not identify the infecting species.

The first report of *Borrelia burgdorferi* s.l. in animals was made by Salinas-Meléndez et al. [82] in DNA obtained from blood of an infected dog. Thus, publications on Lyme disease in Mexico started in the decade between 1990–1999. During this period, the Medical Research Unit of Infectious and Parasitic Diseases of the Mexican Social Security Institute [Unidad de Investigación Médica de Enfermedades Infecciosas y Parasitarias, Centro Médico Nacional SXXI, Instituto Mexicano del Seguro Social] began to study Lyme disease in Mexico. Further studies were done in the states of Nuevo Leon, Sinaloa and Mexico City, that reported human and animal seropositivity throughout the country [81–85].

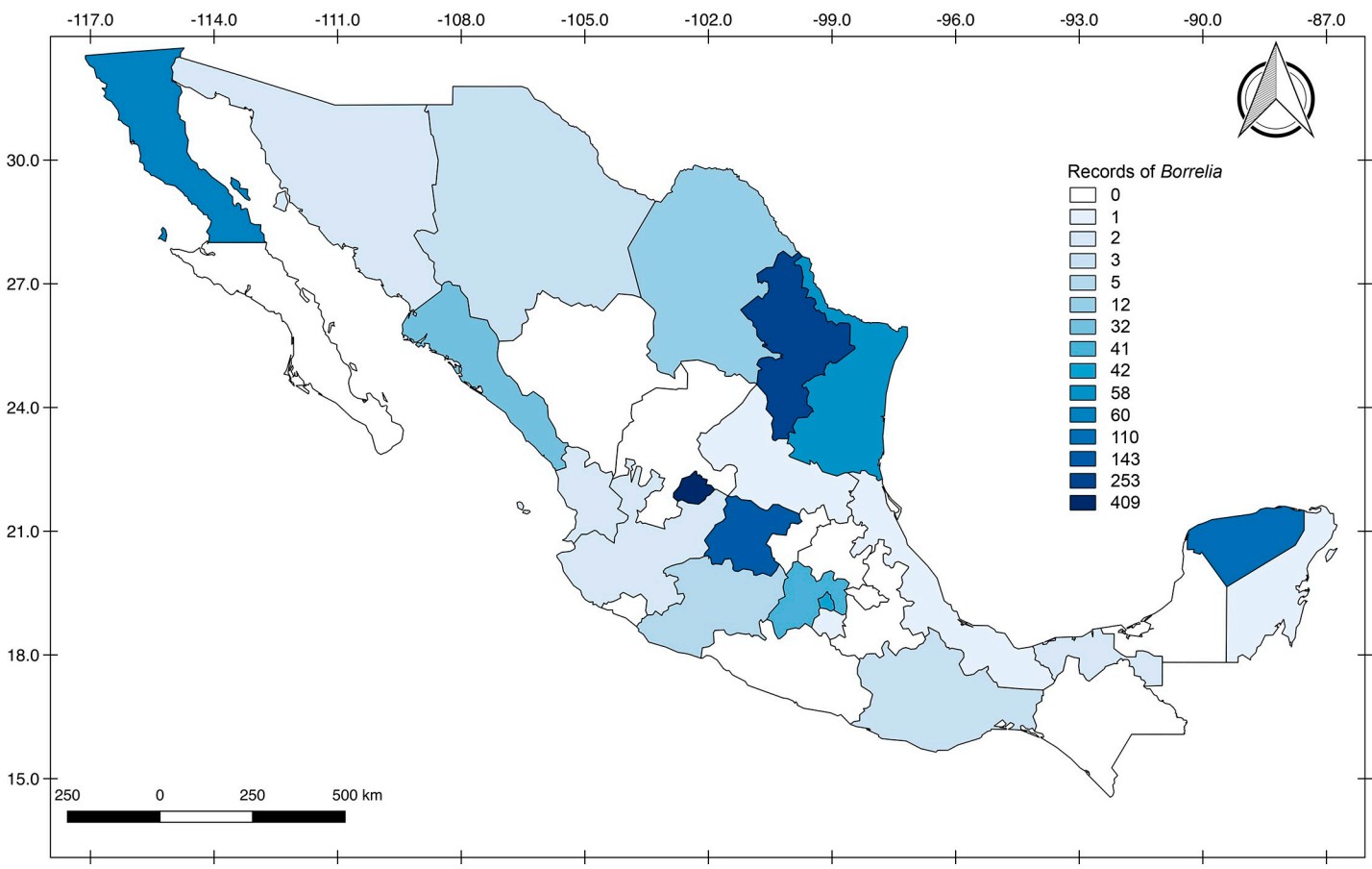

**Fig 2. Geographic distribution of the 1,364 geo-referred records of *Borrelia* in Mexico.** Records include both human cases and animal infections or expositions to *Borrelia* in Mexico from 1939 to 2020.

The introduction of confirmatory tests based on PCR and Western blotting allowed a more accurate diagnosis, as compared to traditional tests (thick blood smear and microscopy) [82, 85]. Yet it was not until the early 2010s, that interest in *Borrelia* research intensified and the numbers of publications increased dramatically. Different research centers began to study the distribution of *Borrelia* in animals and human cases [86–88]. More animal species were screened for *Borrelia* DNA and veterinarians started to perform epidemiological surveys in domestic and wild animals [89, 90].

Thus, studies on Lyme disease in Mexico are recent and focus mainly on comparing LB with the other *Borrelia* groups, both in human cases as well as in animals [6, 88, 91]. Historically, LB has been identified in the majority of human cases (almost 97%) in Mexico, as compared to RF. Yet, considering the drastic increase of human cases of borreliosis in countries where the disease is endemic [19, 92, 93], in Mexico an enhanced effort to diagnose and confirm the infection by species of the LB group still needs to be accomplished. Is important to highlight that human cases of LB are mostly reported in urban communities, however, considering the life cycle of ixodid ticks and the transmission cycle of *B. burgdorferi* s.l., it is unlikely that infections originated in these urban areas, since tick larvae and nymphs feed mainly on wildlife [2, 94]. Therefore, it is more likely that humans become infected after incursions into forest areas, where they are exposed to ticks.

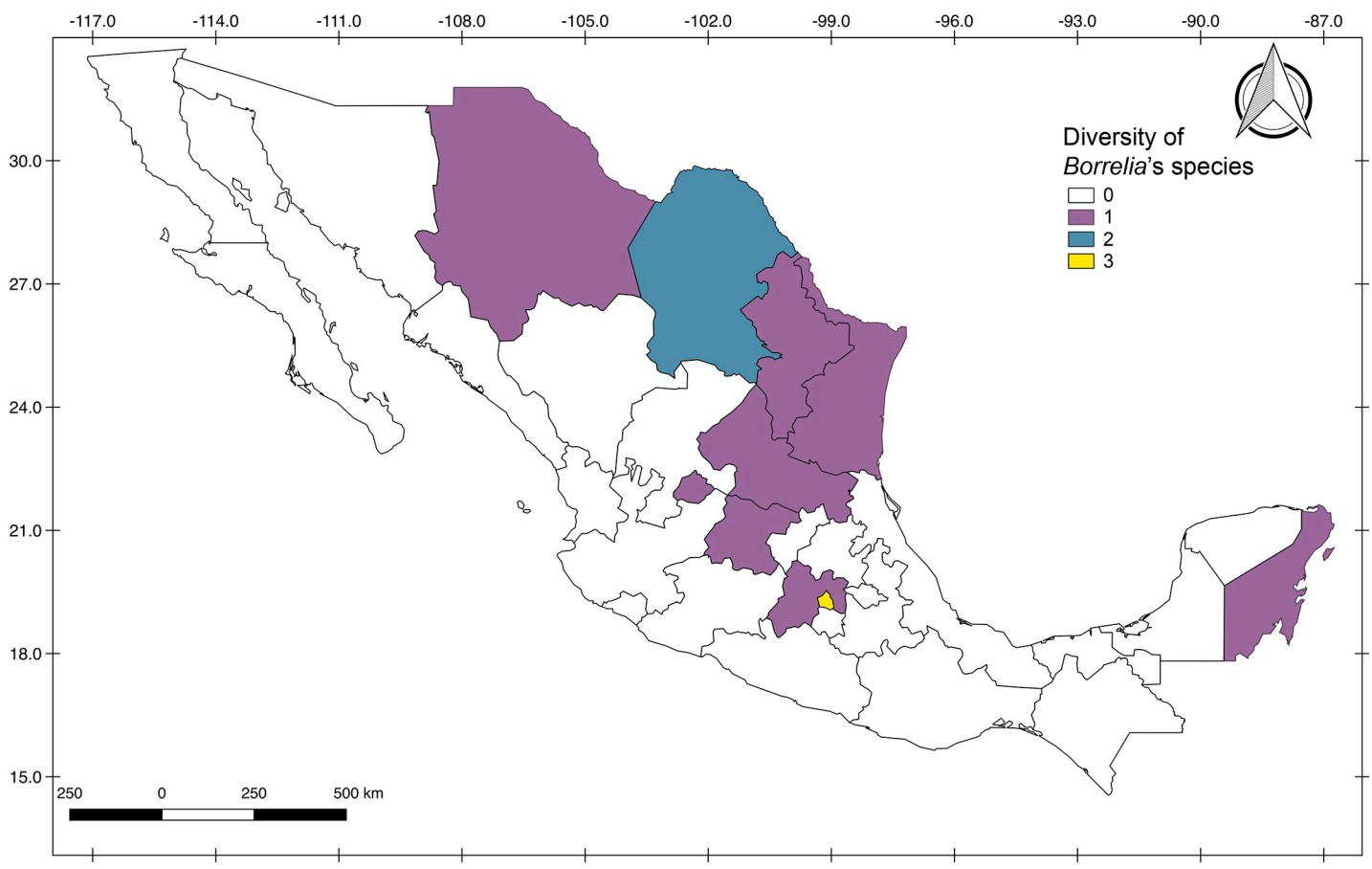

**Fig 3. Geographic distribution of the genus *Borrelia* in Mexico according to species richness.** Colors indicates the number of species of the bacterial genus per State. Records of bacteria defined as *B. burgdorferi* s.l. were not included, since the species was not specified.

Currently, it is not possible to establish the origin of RF infections, although the available data seem to indicate that rural communities are the main origin of the infections, yet further studies are required to show a more complete epidemiological panorama. Since RF is regarded a forgotten and neglected tropical disease [15, 95], and given the scarce number of cases with RF reported by the Mexican Ministry of Health [Sistema Nacional de Información en Salud, Secretaría de Salud], it is very likely that many of the patients that are currently reported as having "fever of unknown origin", lacking identification of the etiological agent, are actually patients with RF [64]. It is therefore imperative to consider RF as a differential diagnosis in these types of cases, especially in rural communities, where most of the cases of RF have occurred in Mexico and where a higher risk of contracting the disease exists, as compared to urban communities [15, 95].

Taken together, the current lack of data on both diseases prevent an accurate epidemiological analysis to be made, nor can risk factors for LB or RF be established in Mexico. However, with the information obtained so far, women appear to be more susceptible to *Borrelia* infections than men, both for LB as for RF. This tendency is similar to data reported in Europe and the US [15, 19, 96].

When analyzing the clinical manifestations reported by patients with suspected Lyme disease in Mexico, including the confirmed case report of this study, the general manifestations include fatigue, fever, arthralgia, paresthesia and myalgias [83, 87, 88, 97, 98]. Chronic Lyme

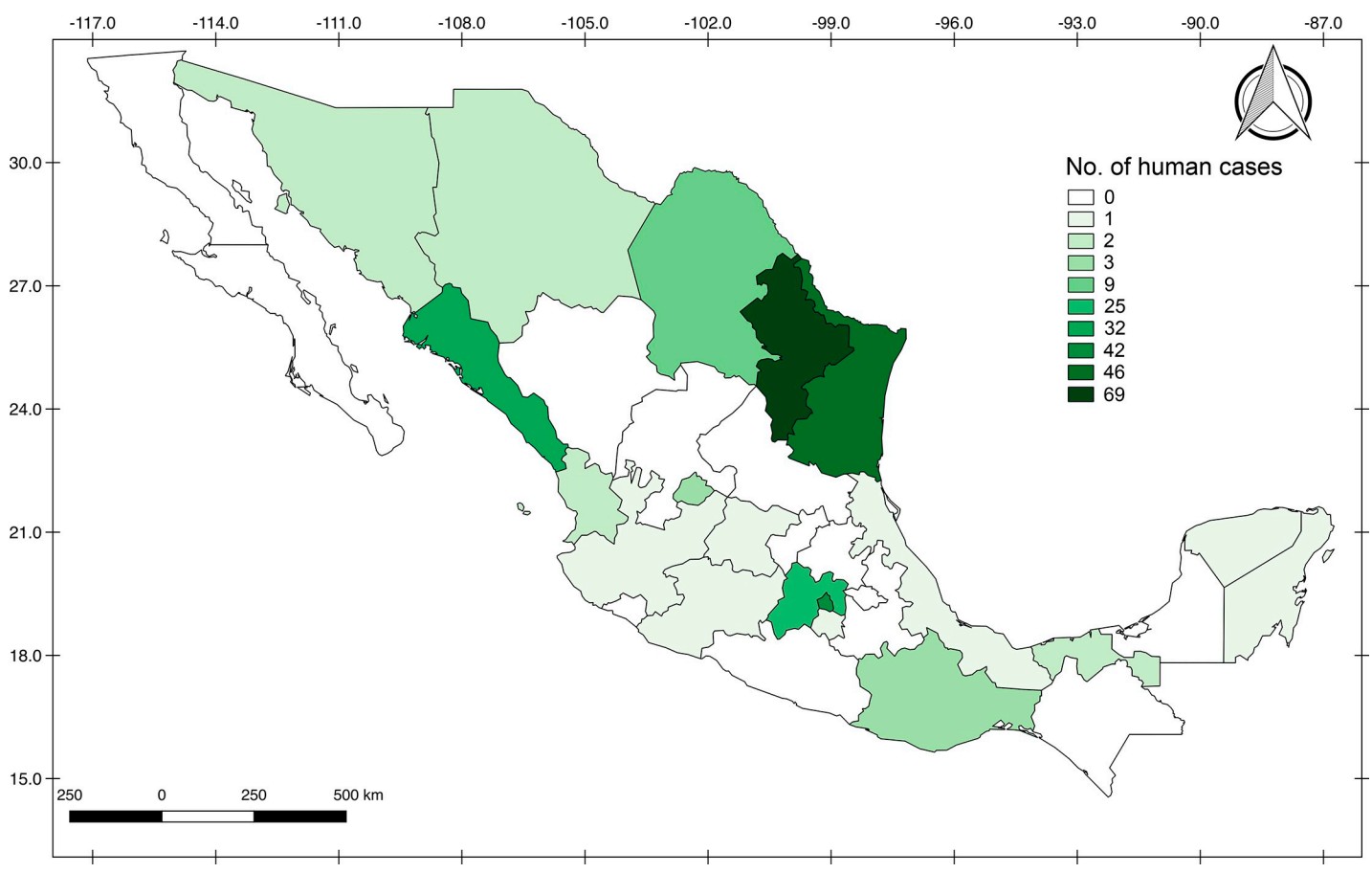

**Fig 4. Geographic distribution of *Borrelia*-infected human cases in Mexico.** Darker color represents higher density.

disease often is accompanied by neurological symptoms of neuroborreliosis, including symmetric paresthesia with ascending distribution in all four limbs, meningo-polyradiculoneuropathy (motor and sensory), arthritis [86, 99] and facial palsy, mainly in children [98]. Dermal lesions include irregular and regular *erythema migrans* with reddish edges, pink center, with a clear mononuclear cell infiltrate in the superficial and deep dermis. Additionally, borrelial lymphocytoma lesions with dense nodular lymphocytic infiltrates in the reticular dermis and well-delineated lymphoid follicles can be found. However skin lesions such as acrodermatitis chronica atrophicans are rare [83, 86, 100]. Taking into account the symptoms of the currently reported case, which include diplopia and bilateral hypoesthesia in both upper and lower limbs, tremors in the arms and short-term memory loss, we suggest that these should also be considered as possible symptoms for Lyme disease in Mexican patients.

On the other hand, clinical manifestations reported for RF in Mexican patients differ from those of Lyme disease, since RF patients show fever paroxysms lasting between 2–7 days, which alternate with periods (4–12 days) of apyrexia. The fevers oscillate between 38.5˚C and 40.8˚C, preceded by intense chills. The most common manifestations are headache, exanthems and weakness (found in three of the cases), as well as splenomegaly, hepatomegaly, diaphoresis, epistaxis and photophobia [80, 101]. Blood counts are characterized by eosinophilia, increased platelet counts and moderate leukocytosis, with 80% polynucleated cells [80].

The analysis of *Borrelia* hosts and vectors in Mexico has shown that most of the *Borrelia* infections in animals have been studied in the order Rodentia, which is considered a potential

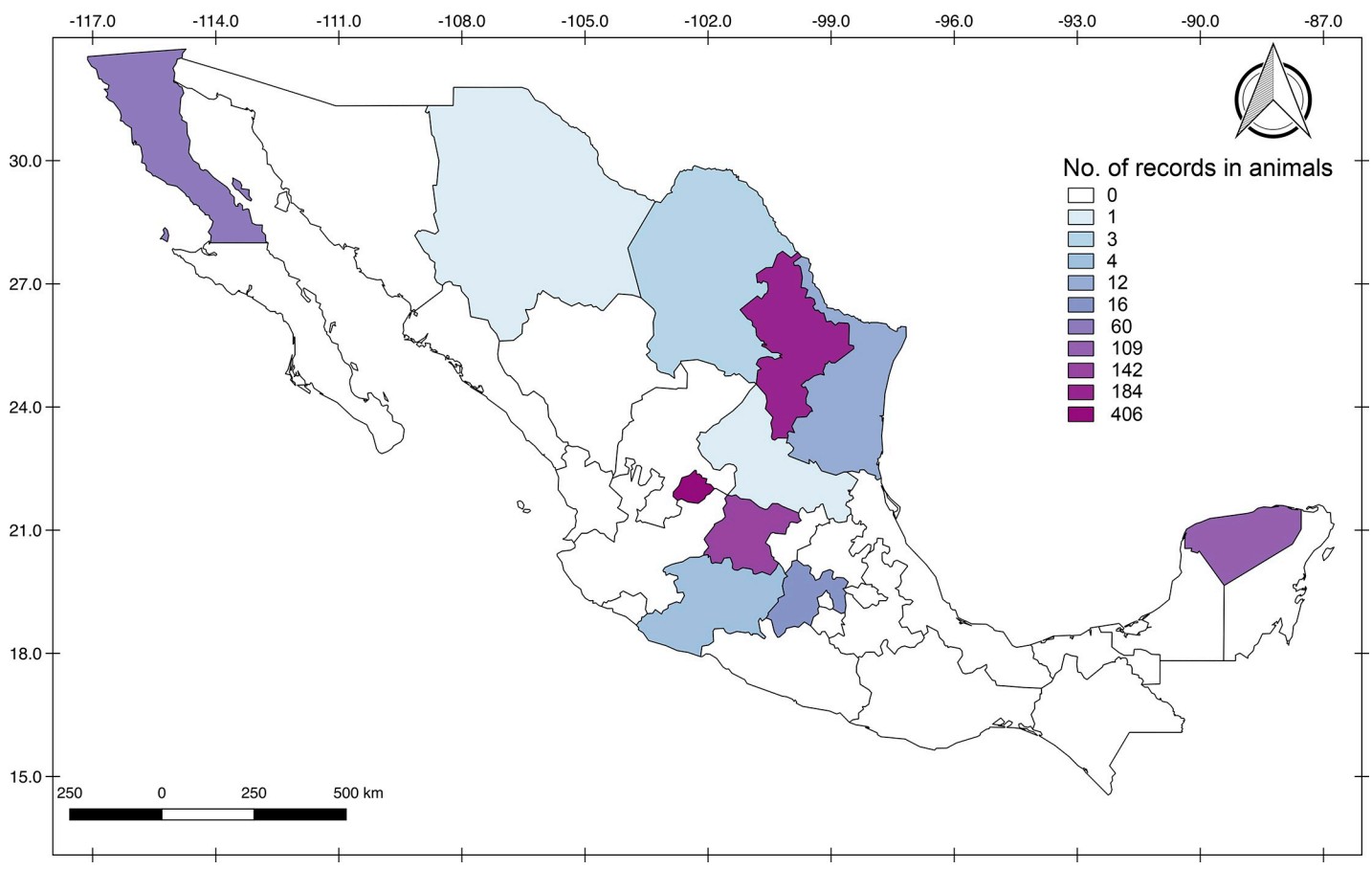

**Fig 5. Geographic distribution of *Borrelia* records in wild animals from Mexico.** Darker color represents higher density.

host for *Borrelia* species. This is in accordance with the literature, where this order has been implicated as one of the most important hosts, for both the LB and the RF groups [2, 15, 19]. Thus, several species of genera *Peromyscus* and *Neotoma* have been recognized as reservoirs for many *Borrelia* species in endemic countries [2, 3, 102, 103]. For the LB species, several mammals (mainly rodents and the white-tailed deer) have also been shown to be hosts in Mexico, in which only three *Borrelia* species have been reported. This low number contrasts with the number of species reported in countries endemic for *Borrelia*, such the USA, where nine species of the LB group have been reported [5, 17, 67, 104].

With regard to vector studies done in Mexico, these show that four of the 26 species of the genus *Ixodes* (*I. affinis*, *I. pacificus*, *I. scapularis* and *I. spinipalpis*) are confirmed vectors of species of the LB group. In contrast, only three vectors have been confirmed for *Borrelia* species of the RF group (*Ornithodors coriaceus*, *Ornithodoros turicatae* and *I. scapularis*), which affect several host species (rodents, deer and dogs). The diversity of RF species (three species) reported in Mexico is low, as compared to the eight species reported in the US [3, 15, 67–69]. This highlights the importance of specifying the tick specimens (mainly those recognized as competent vectors) recovered from various hosts, irrespective of whether they have been recognized as reservoirs. Thus, *I. scapularis*, *I. texanus* and *O. duguesi* specimen retrieved from jaguars, cattle, ringtails and rabbits have been shown to test positive for *Borrelia* (S2 Table). The analysis of diverse wild and domestic animal species is relevant, since they could be

playing a role as maintenance hosts for ticks in the ecosystem, as has been evidenced in white-tailed deer [105, 106].

Taken together, the low host and vector diversity reported in Mexico calls for future studies to update the existing historical records [26, 79, 107]. The important difference in the numbers of registered species between Mexico and the US, an endemic and hotspot of the genus *Borrelia*, is probably due to inadequate sampling in Mexico, but mainly because of the use of serological tests without further confirmation of the bacterial presence. It seems warranted to predict higher numbers of both hosts and vectors in Mexico due to important climatic varieties and diversity of ecosystems that provide ideal habitats and opportunities for sustaining diverse vector and host species of this bacterial genus [67, 70, 78].

Previous reports on the diversity of *Borrelia* species in central and northeastern Mexico show that areas with the highest number of records overlap with those showing high probabilities of occurrence of ticks of the genus *Ixodes* [108]. This observation contrasts with the report of Illioldi-Rangel et al., [108] stating that the state of Durango stands out as having a high density of *Ixodes*, yet no records of *Borrelia* have been reported. Clearly, more studies need to be conducted in Durango to validate potential distribution maps.

In the case of the RF, the existing records are related to the distribution of their vectors [79, 81]. The importance of analyzing potential distribution of vectors is based on the fact that it not only permits to direct sampling efforts, but is also crucial for knowledge on vector-borne pathogens in sub-sampled areas and for optimizing epidemiological surveys [108, 109].

To the best of our knowledge, there are no associations between human cases with those of animals or vectors in Mexico. Although human cases have been reported the states Morelos, Oaxaca, Quintana Roo, Tabasco, Veracruz and Mexico City, animal studies are lacking. This situation has also been shown for another groups of pathogens in Mexico such as viruses [63]. The human cases reported in Mexico City most likely refer to patients that were transferred from regional health centers to larger hospitals in the city for better diagnosis and treatment. Natural transmission of *Borrelia* within Mexico City seems highly unlikely. This phenomenon has also been observed for other diseases, in which diagnostic centers are concentrated in larger cities [110]. Sinaloa and Nuevo Leon have become important states, where most of the human *Borrelia* transmission has been reported in Mexico. Regrettably, many cases have only been reported in local epidemiology reports, in pathology departments of hospitals or clinics, or are submitted to local journals with low accessibility and limited distribution, making the information on *Borrelia* difficult to obtain, thus generating inaccurate data.

Since several international organizations and centers have recognized the diseases caused by *Borrelia* as neglected or as a major health threat [1, 93, 95], this now shows the necessity to establish a surveillance program and a specialized reference research center in Mexico, where isolates and *Borrelia* strains can be collected to facilitate more precise information on these pathogens, as well as to identify more specific antigens that could improve the diagnosis.

We consider that our geographic approach with spatial distribution data will now provide valuable information on *Borrelia* for human and animal health authorities and may also be relevant for future control and prevention programs, as well as a guide to direct capture efforts for specific animal studies. Creating a universal and open access database of all published records for scientists, entomologists and public health authorities, can help establish linkages among groups working with the genus *Borrelia* in Mexico. Even though the study of the genus *Borrelia* in Mexico is currently poorly assessed, new work groups are being formed and existing groups are consolidating, which together will increase the knowledge of this bacterial genus in Mexico and complete the epidemiological panorama.

## Supporting information

**S1 Table. Borreliosis human cases reported in Mexico.**
(DOCX)

**S2 Table. *Borrelia* detection in Mexican animals.**
(DOCX)

## Acknowledgments

This work was supported by Universidad Nacional Autónoma de México (UNAM) with grant number PAPIIT IN211418. Pablo Colunga-Salas is a doctoral student of Programa de Doctorado en Ciencias Biomédicas, Universidad Nacional Autónoma de México (UNAM) and has received the CONACyT fellowship with number 463798.

## Author Contributions

**Conceptualization:** Pablo Colunga-Salas, Sokani Sánchez-Montes.

**Formal analysis:** Pablo Colunga-Salas.

**Funding acquisition:** Ingeborg Becker.

**Methodology:** Pablo Colunga-Salas, Sokani Sánchez-Montes.

**Project administration:** Pablo Colunga-Salas, Ingeborg Becker.

**Resources:** Patricia Volkow, Ingeborg Becker.

**Visualization:** Sokani Sánchez-Montes, Patricia Volkow, Adriana Ruíz-Remigio, Ingeborg Becker.

**Writing – review & editing:** Pablo Colunga-Salas, Sokani Sánchez-Montes, Patricia Volkow, Adriana Ruíz-Remigio, Ingeborg Becker.

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
