## [Decision Letter · Decision Letter 0]

5 May 2020

PONE-D-20-07127

Lyme Borreliosis and Relapsing Fever in humans and wildlife of Mexico

PLOS ONE

Dear Dr Becker,

Thank you very much for submitting your manuscript "Lyme Borreliosis and Relapsing Fever in humans and wildlife of Mexico" (#PONE-D-20-07127) for review by PLOS ONE. As with all papers submitted to the journal, your manuscript was fully evaluated by academic editor (myself) and by independent peer reviewers. The reviewers appreciated the attention to an important health topic, but they raised substantial concerns about the paper that must be addressed before this manuscript can be accurately assessed for meeting the PLOS ONE criteria. Therefore, if you feel these issues can be adequately addressed, we invite you to submit a revised version of the manuscript that addresses the points raised during the review process. We can’t, of course, promise publication at that time.

We would appreciate receiving your revised manuscript by Jun 19 2020 11:59PM. To enhance the reproducibility of your results, we recommend that if applicable you deposit your laboratory protocols in protocols.io, where a protocol can be assigned its own identifier (DOI) such that it can be cited independently in the future. For instructions see: http://journals.plos.org/plosone/s/submission-guidelines#loc-laboratory-protocols

We look forward to receiving your revised manuscript.

Kind regards,

Abdallah M. Samy, PhD

Academic Editor

PLOS ONE

**Journal Requirements:**

'This work was supported by PAPIIT IN211418. Pablo Colunga-Salas is a doctoral student of Pograma de Doctorado en Ciencias Biomédicas, Universidad Nacional Autónoma de México (UNAM) and has received the CONACyT fellowship with number 463798.'

'The funders had no role in study design, data collection and analysis, decision to

publish, or preparation of the manuscript'

**Reviewers' comments:**

Reviewer's Responses to Questions

**Comments to the Author**

1. Is the manuscript technically sound, and do the data support the conclusions?

Reviewer #1: No

Reviewer #2: Yes

Reviewer #3: Partly

Reviewer #4: Yes

Reviewer #5: Yes

2. Has the statistical analysis been performed appropriately and rigorously? 

Reviewer #1: No

Reviewer #2: N/A

Reviewer #3: N/A

Reviewer #4: Yes

Reviewer #5: Yes

3. Have the authors made all data underlying the findings in their manuscript fully available?

Reviewer #1: No

Reviewer #2: Yes

Reviewer #3: Yes

Reviewer #4: Yes

Reviewer #5: Yes

4. Is the manuscript presented in an intelligible fashion and written in standard English?

Reviewer #1: Yes

Reviewer #2: Yes

Reviewer #3: No

Reviewer #4: Yes

Reviewer #5: Yes

5. Review Comments to the Author

Reviewer #1: (1) The term "borreliosis" for both Lyme disease and relapsing fever is not common usage. This will likely confuse readers. Why not just write about Lyme disease (or Lyme borreliosis) and relapsing fever?

(2) The article is mainly a case report with a review of the literature. It is not a research article. Table 1 is more appropriate for a textbook chapter.

(3) The nucleotide sequence with accession number of MN607028 (line 165) is not publicly available. This is a major problem.

(4) In any case, the examples of "B. burgdorferi" in the tree are not from a recognized reference strain, preferably the type strain for the species, but a short fragment of the flagellin gene from Illinois, a state with a low incidence of Lyme disease.

Reviewer #2: The manuscript is well written and the authors were able to present their study well. The compilation of data presenting the epidemiological trend of Lyme Borreliosis and Tick borne Relapsing fever in Mexico will be helpful for further research in this field. However, I would suggest the authors proof read the manuscript for minor errors. For example: Relapsing Fever (RF) is written as FR in Table 1 under the Borrelia group column. Also, the full from of all abbreviations used in the manuscript needs to be given the first time it is mentioned in the paper.

Reviewer #3: General Comments

The manuscript by Colunga-Salas and colleagues aims to "compile all the information published on Borrelia in Mexico and to present actual epidemiological trends." Although this is a noble goal, the epidemiological data appears to be too sparse and fragmented to achieve this goal. The other feature of the report is the detailed genomic description of the first documented endemic case of Borrelia burgdorferi sensu stricto (Bbss) in Mexico. Reorganization of the manuscript to focus on this case would make sense.

Major Comments

1. The detailed genomic description of the first documented endemic case of Bbss in Mexico is certainly original and should be the major focus of the report. The manuscript should include the actual BLAST sequence from that case to confirm its validity. Starting with this case will show where future studies should go in determining the epidemiology of tick-borne disease in Mexico. In addition, a detailed travel history should be given because the patient may have acquired his infection outside of Mexico.

2. The epidemiology of Borrelia burgdorferi (Bb) and relapsing fever Borrelia (RFB) in Mexico is certainly of interest, but the available data was limited to 18 states, thereby introducing significant selection bias into the study results. Presentation of this review data could be done in a more organized and concise format with less speculation about the implications of the incomplete data.

3. A benefit of separating out the case report from the epidemiology is that the case can be presented in a more organized fashion, with clinical features, test methodology, results and conclusions. Right now the clinical and epidemiology aspects are mixed together, and it is a mess.

4. Page 4, line 84: "The competent vectors of this bacterial species are hard ticks of the Ixodes ricinus complex". Ixodes ricinus is mainly found in Europe, so better to say that Bb is transmitted by hard ixodid ticks. The authors should also mention soft argasid ticks that transmit RFB (Fesler et al, Healthcare 2020;8:97).

5. For future review, it would be helpful to separate the Tables from the text. Much easier to follow the bouncing ball that way.

6. Page 15, line 221: Isn't it surprising that Bb was reported mostly in urban communities? The number of recognized RFB cases (5) was too small to draw geographical conclusions.

7. Numerous spelling and grammatical errors need to be corrected. Examples (comments): prompts to include both diseases as possible causes of reports on “fever of unknow origin” (awkward sentence and sp), disperse (sp), incriminated (implicated), widely dispersed (just the opposite: not disseminated), a red skin lesion after visiting the family farm that lasted several weeks (the lesion, not the visit), sensibility (sensitivity), short time (short term), starting performing (?), BL(LB?), sensorial (sensory), sub-sampling (inadequate sampling)

Minor Comments

1. Page 22, line 331: REP and MAB sentence is hard to understand

2. Reference 41 is incomplete

Reviewer #4: General Comments

This study attempted to document published information on Borrelia in Mexico, with reference to risk factors for the human population, the diversity of Borrelia species and their geographic distribution.

The author could improve the manuscript by citing references from the actual work that reported the Borrelia species, rather than utilising single review article (Reference 3 and 10), that does not refer to the work done on the pathogen by initial authors. This is especially relevant to the tables.

Abstract

Line 31

Insert full stop after human and start a new sentence with Women : species were reported in a total of 1,347 reports, of which 398 were of humans. Women and children from rural communities appear to be……

Line 36

only made from 18 states out of how many states ?

Introduction

Table 1. Pathogenic Borrelia species:

The first group on table 1 is RF not FR.

The proper references for Candidatus Borrelia kalaharica should be:

1. Fingerle V, Pritsch et al. 2016. “Candidatus Borrelia kalaharica” detected from a febrile traveller returning to Germany from vacation in southern Africa. PLoS Negl Trop Dis 10:e0004559. https://doi.org/10.1371/journal.pntd.0004559.

2. Cutler S.J., Ahmed A.O., et al, 2018. Ornithodoros savingyi – the tick vector of Candidatus Borrelia kalaharica in Nigeria. Journal of Clinical Microbiology. 56(9): e00532-18. https://jcm.asm.org/content/jcm/56/9/e00532-18.full.pdf

Methods

Line 104: include year of coverage of studies 1939 and 2020

Line103-110 refers to the methodology used for literature search for mammals, human cases and ticks in Mexico. What is line117-123 meant to infer? Are they additional database for human studies?

Line 149: DNA was extracted from? Blood or Sera? Please clarify and include in the sentence

Results

Table 2: Include a first column for LB and RF groups:

Group Borrelia species No. of records Type of host

LB

Discussion

Line 305: Brumpt what year?

Line 308: Pilz and Mooser what year?

Line 315: Salinas-Meléndez et al.include the year

Line 336: Did you mean BL or LB?

The discussion is realistic with findings.

Reviewer #5: In the manuscript entitled Lyme Borreliosis and Relapsing fever if humans and wildlife of Mexico by Colunga-Salas et. al. have gathered information of distribution, diversity and density of Borrelia species in different parts of Mexico. They found in total 17 different mammalian 14 different tick species hosting borrelia spirochetes. Authors also describe the first endemic case of Lyme borreliosis in Mexico verified by sequence analysis of the causative strain. The manuscript is interesting and increases knowledge of very important pathogens, Lyme disease and relapsing fever borrelia. There are very few points which might need clarification.

In the table 1 the authors have gathered information using one manual of systematics and some case studies and articles. Is the list trying to be exhaustive? At least I know some species, e.g. Borrelia finlandensis, which according to some references should be a species of its own and it is not mentioned in the table. Thus, a wider description of how species were chosen and if the list is covering all pathogenic species might be useful addition. Also in the introduction it might be good to tell the readers about latest development in Borrelia -systematics, e.g, division of the genus into Borrelia and Borreliella. Also term Borreliella might be useful to add as keyword in database search.

As not all readers are familiar with zoology, it might be aid readability and be educational if existing common names of animal species could be added to the table 3.

In the map figures the figure legends and numbers in the actual figures are not very informative. It might be not clear to reader what is meant “records of borrelia” for example in the figure 2. Does it mean “amount of borrelia findings from humans and animals isolated during 20xx-20xx” or something else? The figure legends should be carefully checked and more informative.

In the beginning of discussion it would be interesting to know, when the first borrelia findings were made.

Minor points:

- Instead of “sex” maybe gender could be used (page 15, line 213)

6. PLOS authors have the option to publish the peer review history of their article (what does this mean?). If published, this will include your full peer review and any attached files.

Reviewer #1: No

Reviewer #2: Yes: Akhila Poruri

Reviewer #3: No

Reviewer #4: No

Reviewer #5: No

---

## [Author Response · Author response to Decision Letter 0]

23 Jun 2020

Response to Reviewers

We thank the reviewers for their comments and have now addressed all their points making the pertinent changes throughout the manuscript. The detailed response are included below. 

REVIEWER #1

(1) The term "borreliosis" for both Lyme disease and relapsing fever is not common usage. This will likely confuse readers. Why not just write about Lyme disease (or Lyme borreliosis) and relapsing fever? 

Response: We agree with the reviewer and changed the term “borreliosis” in the abstract. Additionally, we proposed changes in the title and short title, and made changes throughout the manuscript (Lines 63, 108, 112, 243, 248, 363, 420, 470, 490).

(2) The article is mainly a case report with a review of the literature. It is not a research article. Table 1 is more appropriate for a textbook chapter.

Response: One of the aims of the study was to assess the epidemiological situation of borreliosis in Mexico, since most of the studies done so far are not easily available. An epidemiological update is needed to incentivize studies on Lyme borreliosis and Relapsing fever and to consider Relapsing fever as an important differential diagnosis in cases of “Fever of Unknown Origin” in Mexico.

Regarding Table 1, we consider it important to update knowledge on species that are pathogenic for both humans and animals, for readers not familiar with the genus Borrelia.

(3) The nucleotide sequence with accession number of MN607028 (line 165) is not publicly available. This is a major problem.

Response: We are aware of this situation, however, when submitting the sequence to GenBank, it was requested that it be released in 2022 or earlier, if the accession number was published. Despite this inconvenience, the publisher asked us for the sequence, which we shared. Since we don’t know whether this document was shared with the reviewer, we include the sequence:

>MexicanPatient

TTCCGATGCAGACAGAGGTTCTATACAAATTGAAATAGAGCAACTTACAGACGAAATTAATAGAATTGCTGATCAAGCTCAATATAACCAAATGCACATGTTATCAAACAAATCTGCTTCTCAAAATGTAAGAACAGCTGAAGAGCTTGGAATGCAGCCTGCAAAAATTAACACACCAGCATCACTTTCAGGGTCTCAAGCGTCTTGGACTTTAAGAGTTCATGTTGGAGCAAACCAAGATGAAGCTATTGCTGT

(4) In any case, the examples of "B. burgdorferi" in the tree are not from a recognized reference strain, preferably the type strain for the species, but a short fragment of the flagellin gene from Illinois, a state with a low incidence of Lyme disease.

Response: We thank the reviewer and have re-analyzed the set of sequences. Three B31 sequences were included (Figure 1). Accession numbers: AE000783 and CP019767. 

Furthermore, changes were made in methods and figure legend 1, since new substitution model analyses was required, and a different substitution model was obtained from the new alignment (Lines 163-164, 168, 176).

REVIEWER #2

The manuscript is well written, and the authors were able to present their study well. The compilation of data presenting the epidemiological trend of Lyme Borreliosis and Tick borne Relapsing fever in Mexico will be helpful for further research in this field. However, I would suggest the authors proof read the manuscript for minor errors. For example: Relapsing Fever (RF) is written as FR in Table 1 under the Borrelia group column. Also, the full from of all abbreviations used in the manuscript needs to be given the first time it is mentioned in the paper.

Response: We thank the reviewer and have carefully revised the abbreviations throughout the manuscript and made sure that the first time any abbreviation was used, it was fully written. Changes were done throughout the manuscript (Lines 162-163, 215-216, 237). We also corrected the error (changing FR to RF) in Table 1.

REVIEWER #3 

(1) General Comments

The manuscript by Colunga-Salas and colleagues aims to "compile all the information published on Borrelia in Mexico and to present actual epidemiological trends." Although this is a noble goal, the epidemiological data appears to be too sparse and fragmented to achieve this goal. The other feature of the report is the detailed genomic description of the first documented endemic case of Borrelia burgdorferi sensu stricto (Bbss) in Mexico. Reorganization of the manuscript to focus on this case would make sense.

Response: We appreciate the recommendation and have now re-organized the manuscript into sections (A), related to the case report and section (B) related to review of the literature. The case report is now separated and fully presented after the introduction (Lines 114-180).

Regarding the epidemiological panorama, we are aware of the limited data available, however, we consider that an actualized panorama alerts on this gap of knowledge and helps to incentivize future research to bring data on human infections up to date in sites where animal infections have been recorded, and vice versa. Furthermore, our effort has now evidenced that it is important to consider Relapsing Fever as a differential diagnosis in cases currently diagnosed as “Fever of Unknown Origin” in Mexico.

(2) Major Comments:

1. The detailed genomic description of the first documented endemic case of Bbss in Mexico is certainly original and should be the major focus of the report. The manuscript should include the actual BLAST sequence from that case to confirm its validity. Starting with this case will show where future studies should go in determining the epidemiology of tick-borne disease in Mexico. In addition, a detailed travel history should be given because the patient may have acquired his infection outside of Mexico.

Response: We now include travel information, as detailed as the patient reported it (Lines 119-123). We were careful to assess whether he had made trips outside of Mexico, yet he only reported local trips within the state of Veracruz. We agree that this case sheds some light as to where future epidemiological studies should go and which can be enriched by the most complete information to date on the current epidemiological panorama of Borreliosis in Mexico, that we now present.

We now include the BLAST sequence in the manuscript to confirm the validity of the case (Lines 168-173

(3) The epidemiology of Borrelia burgdorferi (Bb) and relapsing fever Borrelia (RFB) in Mexico is certainly of interest, but the available data was limited to 18 states, thereby introducing significant selection bias into the study results. Presentation of this review data could be done in a more organized and concise format with less speculation about the implications of the incomplete data.

Response: We thank the reviewer and have now re-organized the manuscript into sections (A), related to the case report and section (B) related to review of the literature. The case report is now separated and fully presented after the introduction (Lines 114-180).

The epidemiological panorama is more organized, and less speculation is made about the incomplete epidemiological panorama of borreliosis in Mexico. We also discuss this point (Lines 374-379, 380-383, 446)

(4) A benefit of separating out the case report from the epidemiology is that the case can be presented in a more organized fashion, with clinical features, test methodology, results and conclusions. Right now the clinical and epidemiology aspects are mixed together, and it is a mess.

Response: After the reviewer's assertive comment of placing the case report as the center of attention, we consider that the case report and the epidemiological panorama of borreliosis are now more understandable.

(5) Page 4, line 84: "The competent vectors of this bacterial species are hard ticks of the Ixodes ricinus complex". Ixodes ricinus is mainly found in Europe, so better to say that Bb is transmitted by hard ixodid ticks. The authors should also mention soft argasid ticks that transmit RFB (Fesler et al, Healthcare 2020;8:97).

Response: We made the pertinent change about the Ixodes ricinus complex (Lines 94-95). Concerning to RF vectors, the information about that soft ticks are the main vectors are present in Lines 85-87.

(6) For future review, it would be helpful to separate the Tables from the text. Much easier to follow the bouncing ball that way.

Response: We agree, however, the format of PLoS ONE indicates that tables must be placed next to the paragraph where they were referred for the first time.

(7) Page 15, line 221: Isn't it surprising that Bb was reported mostly in urban communities? The number of recognized RFB cases (5) was too small to draw geographical conclusions.

Response: We agree that is surprising that LD cases were reported mostly in urban communities. On the other hand, as the reviewer says, the number of RF cases is too small, for this reason, we discuss the most likely reason of this situation (Lines 380-383).

(8) Numerous spelling and grammatical errors need to be corrected. Examples (comments): prompts to include both diseases as possible causes of reports on “fever of unknow origin” (awkward sentence and sp), disperse (sp), incriminated (implicated), widely dispersed (just the opposite: not disseminated), a red skin lesion after visiting the family farm that lasted several weeks (the lesion, not the visit), sensibility (sensitivity), short time (short term), starting performing (?), BL(LB?), sensorial (sensory), sub-sampling (inadequate sampling).

Response: We made the pertinent grammatical and spelling changes as the reviewer suggested (Lines 51, 85, 87, 105, 126, 135, 369, 373, 401, 422, 446, 492-493).

Minor Comments:

(9) Page 22, line 331: REP and MAB sentence is hard to understand

Response: We eliminated the clarification in parenthesis, to make it easier to read (Line 369).

(10) Reference 41 is incomplete

Response: We apologize and completed the reference 67 (Lines 694-696).

75. Guzmán-Cornejo C, Robbins RG. The genus Ixodes (Acari: Ixodidae) in Mexico: adult identification keys, diagnoses, hosts, and distribution. Rev Mex Biodivers. 2010;80: 289–298.

REVIEWER #4

(1) The author could improve the manuscript by citing references from the actual work that reported the Borrelia species, rather than utilising single review article (Reference 3 and 10), that does not refer to the work done on the pathogen by initial authors. This is especially relevant to the tables.

Response: We thank the reviewer and now placed the references of the original descriptions for each species (Table 1).

(2) Abstract

Line 31

Insert full stop after human and start a new sentence with Women: species were reported in a total of 1,347 reports, of which 398 were of humans. Women and children from rural communities appear to be…

Response: We made the suggested changes (Line 34).

(3) Line 36, only made from 18 states out of how many states?

Response: We specified the total number of states (Line 39).

(4) Introduction

Table 1. Pathogenic Borrelia species:

The first group on table 1 is RF not FR.

Response: We apologize and corrected the error on Table 1.

(5) On the Relapsing fever transmitted by ticks in Transcaucasia

The proper references for Candidatus Borrelia kalaharica should be:

1. Fingerle V, Pritsch et al. 2016. “Candidatus Borrelia kalaharica” detected from a febrile traveller returning to Germany from vacation in southern Africa. PLoS Negl Trop Dis 10:e0004559. https://doi.org/10.1371/journal.pntd.0004559.

2. Cutler S.J., Ahmed A.O., et al, 2018. Ornithodoros savingyi – the tick vector of Candidatus Borrelia kalaharica in Nigeria. Journal of Clinical Microbiology. 56(9): e00532-18. https://jcm.asm.org/content/jcm/56/9/e00532-18.full.pdf

Response: We thank the reviewer and now include both references (Table 1).

(6) Methods

Line 104: include year of coverage of studies 1939 and 2020

Response: We made the pertinent changes (Lines 184-185).

(7) Line103-110 refers to the methodology used for literature search for mammals, human cases and ticks in Mexico. What is line117-123 meant to infer? Are they additional database for human studies?

Response: Since notification of borreliosis is not mandatory in Mexico, some published case reports and serological surveys of the human Mexican population has been published without being included in the official databases. The opposite also holds true, where some hospitals and clinics voluntarily notify cases of Lyme disease, yet this information is not necessarily published in scientific journals and therefore do not include symptoms nor detailed information. For this reason, we only included them in the geographic analyses. To make clarify this situation, we included this statement in the manuscript (Lines 197-200, 204-205).

(8) Line 149: DNA was extracted from? Blood or Sera? Please clarify and include in the sentence

Response: We made the clarification (Lines 140).

(9) Results

Table 2: Include a first column for LB and RF groups:

Group Borrelia species No. of records Type of host

LB

Response: We accept and made the pertinent changes (Table 2).

(10) Discussion

Line 305: Brumpt what year?

Line 308: Pilz and Mooser what year?

Line 315: Salinas-Meléndez et al.include the year

Response: We followed the instructions of the journal reference style that only cites the author(s), without including the year.

(11) Line 336: Did you mean BL or LB?

Response: We apologize for the error. The correct abbreviation must be LB, which is now changed (Line 373).

(12) The discussion is realistic with findings.

Response: We really appreciate your comment

REVIEWER #5

There are very few points which might need clarification.

(1) In the table 1 the authors have gathered information using one manual of systematics and some case studies and articles. Is the list trying to be exhaustive? At least I know some species, e.g. Borrelia finlandensis, which according to some references should be a species of its own and it is not mentioned in the table. Thus, a wider description of how species were chosen and if the list is covering all pathogenic species might be useful addition. Also in the introduction it might be good to tell the readers about latest development in Borrelia -systematics, e.g, division of the genus into Borrelia and Borreliella. Also term Borreliella might be useful to add as keyword in database search.

Response: In Table 1 we did not intend to report an exhaustive search on all Borrelia species of the literature. The intention was to only include Borrelia species that are pathogenic for humans and/or animals, and therefore many species are not listed. We specified this in the text (Lines 87-95) stating that 25 of the RF group and 11 of the LB group were capable of causing diseases in human or animal populations. To avoid confusion, we change and referred to the original description articles, where the pathogenic species were firstly described, according to Reviewer 4.

We appreciate the recommendation to inform to the reader on the latest taxonomic updates of this genus. We therefore included a paragraph in the introduction regarding this point (Lines 77-84). Additionally, we followed the recommendation and included “Borreliella” as a keyword.

(2) As not all readers are familiar with zoology, it might be aid readability and be educational if existing common names of animal species could be added to the table 3.

Response: We are grateful for your recommendation. We now included English and Spanish common names in Table 3.

(3) In the map figures the figure legends and numbers in the actual figures are not very informative. It might be not clear to reader what is meant “records of borrelia” for example in the figure 2. Does it mean “amount of borrelia findings from humans and animals isolated during 20xx-20xx” or something else? The figure legends should be carefully checked and more informative.

Response: We reviewed all Figure legends, and modified those of Figures 2 and 3.

(4) In the beginning of discussion it would be interesting to know, when the first borrelia findings were made.

Response: The first record of Borrelia in Mexico, as well as the first human case and animal exposure are detailed in the first paragraph of the Discussion. Possibly the misunderstanding arose because we erroneously stated “case” instead if “record”, which was now modified (Line 342). 

(5) Minor points:

- Instead of “sex” maybe gender could be used (page 15, line 213)

Response: We replaced “sex” by “gender” (Line 249).

---

## [Decision Letter · Decision Letter 1]

16 Jul 2020

PONE-D-20-07127R1

Lyme disease and Relapsing fever in Mexico: an overview of human and wildlife infections

PLOS ONE

Dear Dr. Becker,

Thank you very much for submitting your manuscript "Lyme disease and Relapsing fever in Mexico: an overview of human and wildlife infections" for consideration at PLOS ONE. As with all papers reviewed by the journal, your manuscript was reviewed by members of the editorial board and by several independent reviewers. In light of the reviews (below this email), we would like to invite the resubmission of a significantly-revised version that takes into account the reviewers' comments.

We look forward to receiving your revised manuscript.

Kind regards,

Abdallah M. Samy, PhD

Academic Editor

PLOS ONE

**Journal Requirements:**

**Reviewers' comments:**

Reviewer's Responses to Questions

**Comments to the Author**

1. If the authors have adequately addressed your comments raised in a previous round of review and you feel that this manuscript is now acceptable for publication, you may indicate that here to bypass the “Comments to the Author” section, enter your conflict of interest statement in the “Confidential to Editor” section, and submit your "Accept" recommendation.

Reviewer #3: (No Response)

2. Is the manuscript technically sound, and do the data support the conclusions?

Reviewer #3: Yes

3. Has the statistical analysis been performed appropriately and rigorously? 

Reviewer #3: N/A

4. Have the authors made all data underlying the findings in their manuscript fully available?

Reviewer #3: Yes

5. Is the manuscript presented in an intelligible fashion and written in standard English?

Reviewer #3: Yes

6. Review Comments to the Author

Reviewer #3: The authors have done an excellent job of revising the manuscript as suggested by the reviewers. Several remaining spelling and grammatical errors need to be corrected:

1. Lines 28-30: "The aims of this study were (1) to present the first confirmatory evidence of an endemic case of Lyme disease in Mexico and (2) to analyze the epidemiological trend of these xxx diseases...."

2. Lines 35, 61: "...were shown to be..."

3. Line 40: "...of performing..."

4. Lines 42-43: "Finally, the search for Borrelia...is recommended..."

5. Line 115: "in August 2017"

6. Line 416: "apyrexia"

7. Line 418: "exanthems"

8. Line 422: "The analysis of Borrelia...has shown..."

9. Line 437: "... which affect several host species (rodents, deer and dogs)."

10. The Case Report heading on Line 114 should have a subheading of "Clinical Summary and Test Procedures" to match the "Results" subheading on Line 157. Line 132 should be a new paragraph.

11. In addition, the authors should add one recent reference that supports their observations:

Fesler MC, Shah JS, Middelveen MJ, Burrascano JJ, Stricker RB. Lyme disease: diversity of Borrelia species in California and Mexico based on a novel immunoblot assay. Healthcare (Basel) 2020;8:97.

7. PLOS authors have the option to publish the peer review history of their article (what does this mean?). If published, this will include your full peer review and any attached files.

Reviewer #3: **Yes: **Raphael B. Stricker, MD

---

## [Author Response · Author response to Decision Letter 1]

29 Jul 2020

Response to Reviewers

We thank reviewer #3 for the comments and have addressed all the points making the pertinent changes throughout the manuscript. The detailed responses are included below. 

REVIEWER #3

General comment

The authors have done an excellent job of revising the manuscript as suggested by the reviewers. Several remaining spelling and grammatical errors need to be corrected.

Response: We are very thankful with your comment.

1. Lines 28-30: "The aims of this study were (1) to present the first confirmatory evidence of an endemic case of Lyme disease in Mexico and (2) to analyze the epidemiological trend of these xxx diseases...."

Response: We made the pertinent changes (Lines 28-29).

2. Lines 35, 61: "...were shown to be..."

Response: We accept and made the changes (Lines 35, 61).

3. Line 40: "...of performing..."

Response: We made the pertinent change (Line 40).

4. Lines 42-43: "Finally, the search for Borrelia...is recommended..."

Response: We modify the sentence as proposed by the reviewer (Line 44).

5. Line 115: "in August 2017"

Response: We made the pertinent change (Line 117).

6. Line 416: "apyrexia"

Response: We accept and made the change (Line 415).

7. Line 418: "exanthems"

Response: We corrected the spelling (Line 417).

8. Line 422: "The analysis of Borrelia...has shown..."

Response: We made the pertinent change (Line 421).

9. Line 437: "... which affect several host species (rodents, deer and dogs)."

Response: We accepted and made the pertinent changes (Line 436).

10. The Case Report heading on Line 114 should have a subheading of "Clinical Summary and Test Procedures" to match the "Results" subheading on Line 157. Line 132 should be a new paragraph.

Response: We agree with the suggestions of the reviewer and made the pertinent changes (Lines 115, 135).

11. In addition, the authors should add one recent reference that supports their observations:

Fesler MC, Shah JS, Middelveen MJ, Burrascano JJ, Stricker RB. Lyme disease: diversity of Borrelia species in California and Mexico based on a novel immunoblot assay. Healthcare (Basel) 2020; 8:97.

Response: We were aware of this publication. However, we had decided not to include it because the study was not done in Mexico. We only included studies done in Mexico, as was established in the methods (Lines 184-186). Additionally, since the study did not report the place from where the Mexican patients had come, they could not be included in the geographical analysis of our work. Both these points were the reasons for not having considered this reference in our current study.

---

## [Editor Report · Decision Letter 2]

19 Aug 2020

Lyme disease and Relapsing fever in Mexico: an overview of human and wildlife infections

PONE-D-20-07127R2

Dear Dr. Becker,

We’re pleased to inform you that your manuscript has been judged scientifically suitable for publication and will be formally accepted for publication once it meets all outstanding technical requirements.

Kind regards,

Abdallah M. Samy, PhD

Academic Editor

PLOS ONE

---

## [Editor Report · Acceptance letter]

9 Sep 2020

PONE-D-20-07127R2 

Lyme disease and Relapsing fever in Mexico: an overview of human and wildlife infections 

Dear Dr. Becker:

I'm pleased to inform you that your manuscript has been deemed suitable for publication in PLOS ONE. Congratulations! Your manuscript is now with our production department. 

Kind regards, 

on behalf of

Dr. Abdallah M. Samy 

Academic Editor

PLOS ONE